# BATF relieves hepatic steatosis by inhibiting PD1 and promoting energy metabolism

**Zhiwang Zhang[1†], Qichao Liao[1†], Tingli Pan[1], Lin Yu[1], Zupeng Luo[1], Songtao Su[1], Shi Liu[1], Menglong Hou[1], Yixing Li[2], Turtushikh Damba[3], Yunxiao Liang[1], Lei Zhou[1]\***

[1]Institute of Digestive Disease, Guangxi Academy of Medical Sciences, the People's Hospital of Guangxi Zhuang Autonomous Region, Nanning, China; [2]College of Animal Science and Technology, Guangxi University, Nanning, China; [3]School of Pharmacy, Mongolian National University of Medical Sciences, Ulan Bator, Mongolia

**\*For correspondence:**
zhoulei@gxu.edu.cn

[†]These authors contributed equally to this work

**Competing interest:** The authors declare that no competing interests exist.

**Abstract** The rising prevalence of nonalcoholic fatty liver disease (NAFLD) has become a global health threat that needs to be addressed urgently. Basic leucine zipper ATF-like transcription factor (BATF) is commonly thought to be involved in immunity, but its effect on lipid metabolism is not clear. Here, we investigated the function of BATF in hepatic lipid metabolism. BATF alleviated high-fat diet (HFD)-induced hepatic steatosis and inhibited elevated programmed cell death protein (PD)1 expression induced by HFD. A mechanistic study confirmed that BATF regulated fat accumulation by inhibiting PD1 expression and promoting energy metabolism. PD1 antibodies alleviated hepatic lipid deposition. In conclusion, we identified the regulatory role of BATF in hepatic lipid metabolism and that PD1 is a target for alleviation of NAFLD. This study provides new insights into the relationship between BATF, PD1, and NAFLD.

## eLife assessment

This **valuable** study presents reports on the role of the transcription factor BATF and its target PD1 in lipid metabolism including a model of nonalcoholic fatty liver disease (NAFLD). Overall, the evidence supporting the conclusions is **convincing**. The work will be of interest to medical biologists working on NAFLD.

## Introduction

Nonalcoholic fatty liver disease (NAFLD) has become a prevalent chronic liver disease that threatens human health globally. Studies have shown that the global prevalence of NAFLD is about 25% and this trend is still rising (*Younossi et al., 2016*). The clinical diagnostic criterion for NAFLD is that >5% of the liver has steatosis (*Siddiqui et al., 2018*). Further deterioration of NAFLD leads to nonalcoholic steatohepatitis (NASH) (*Pierantonelli and Svegliati-Baroni, 2019*) and liver fibrosis, markedly increased risk of adverse outcomes including overall mortality, and liver-specific morbidity and mortality, respectively (*Cotter and Rinella, 2020*).

Transcription factors (TFs) are a class of DNA-binding proteins whose gene regulation ability is critical to the molecular state of cells (*Simicevic and Deplancke, 2017*). Multiple studies have shown that TFs are closely related to NAFLD. The currently discovered TFs related to liver lipid metabolism include peroxisome proliferator-activated receptors (PPARs; *Gross et al., 2017*), liver X receptors (LXRs; *Wang et al., 2015*), and sterol-regulatory element binding proteins (SREBPs; *Eberlé et al., 2004*). These TFs are essential for maintaining the body's lipid metabolism balance through the

regulation of lipid metabolism genes (*Karagianni and Talianidis, 2015*). Drugs targeting TFs have been developed, and are widely used in the prevention and treatment of lipid metabolic disorders such as obesity, hyperlipidemia, diabetes, etc., and show excellent clinical effects and potential applications (*Uyeda and Repa, 2006*; *Mota de et al., 2017*).

Basic leucine zipper ATF-like transcription factor (BATF), BATF2 and BATF3 belong to the activator protein (AP)–1 family (*Sopel et al., 2016*). Initially, BATF family members were only considered to be inhibitors of AP-1-driven transcription, but recent studies have found that these TFs have unique transcriptional activities in dendritic cells, B cells, and T cells (*Murphy et al., 2013*). This indicates that their functions may not be fully understood. Currently, BATF has not been found to be directly related to lipid metabolism. The purpose of this study was to explore this relationship. Our study provides a new perspective for understanding the occurrence of NAFLD.

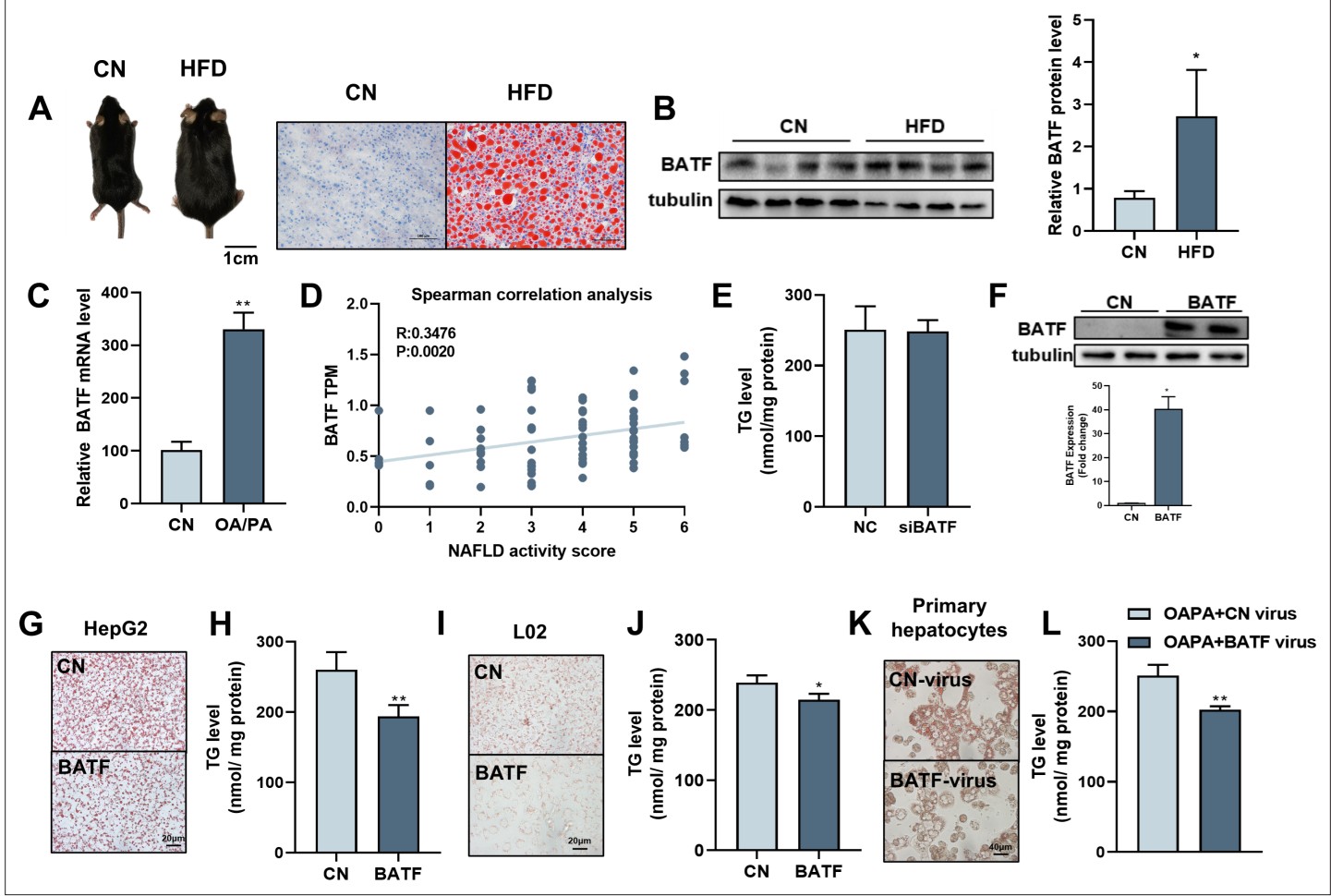

**Figure 1.** Effects of BATF on lipid deposition in hepatocytes under high-fat diet. (**A**) The mice and liver Oil red O staining in normal diet group (CN) and high-fat diet group (HFD). Bar, 1 cm (left panel) and 100 µm (right panel). (**B**) The protein expression of BATF in liver tissues (n=4). (**C**) The mRNA expression of BATF in liver tissues (n=5). (**D**) Spearman correlation Analysis between TPM of BATF and NAFLD Patients with Different Degrees (n=4–18). (**E**) Triglyceride content (n=5). (**F**) Detection of BATF overexpression in HepG2 (n=2). (**G**) Oil red O staining of HepG2 cells with OA/PA when BATF was overexpressed and (**H**) triglyceride content (n=4). (**I**) Oil red O staining of L02 cells with OA/PA when BATF was overexpressed and (**J**) triglyceride content (n=3). (**K**) Oil red O staining of primary hepatocytes with OA/PA when BATF was overexpressed and (**L**) triglyceride content (n=3). The data are expressed as mean ± SD. *p<0.05, **p<0.01.

The online version of this article includes the following source data and figure supplement(s) for figure 1:

**Source data 1.** Effects of BATF on lipid deposition in hepatocytes under high-fat diet.

**Figure supplement 1.** Effects of BATF on lipid levels in hepatocytes.

**Figure supplement 1—source data 1.** Effects of BATF on lipid levels in hepatocytes.

## Results

### BATF reduces high-fat-diet-induced lipid accumulation in hepatocytes

High-fat diet (HFD) promotes hepatic lipid accumulation and leads to dysregulation of lipid metabolism. We detected hepatic BATF expression in mice fed an HFD for 14 weeks to investigate the role of BATF in hepatic lipid metabolism. HFD significantly increased liver lipid deposition and BATF levels in mice (*Figure 1A and B*). HepG2 cells were treated with oleic acid/palmitic acid (OA/PA) to mimic HFD. It promoted triglyceride (TG) accumulation (*Figure 1—figure supplement 1A*) and increased BATF expression (*Figure 1C*). Our analysis of data from patients with NAFLD showed that BATF levels increased with increasing NAFLD score (*Figure 1D*). We inhibited BATF in HepG2 cells (*Figure 1— figure supplement 1B*) and found no effect on cellular TG accumulation (*Figure 1E*). This indicated that inhibition of BATF could not regulate lipid deposition in hepatocytes (*Figure 1F*). Therefore, the role of BATF overexpression was studied. Oil red O staining on lipid droplets and TG detection showed that BATF alleviated OA/PA-induced HepG2 cellular fat deposition (*Figure 1G and H*). BATF also reduced TG accumulation in L02 human hepatocytes (*Figure 1I and J*). To confirm the results, we isolated mouse primary hepatocytes and infected them with a virus expressing BATF. The results also showed that BATF alleviated accumulation of lipid droplets and TG (*Figure 1K and L*). In AML12 cells, we found consistent results (*Figure 1—figure supplement 1C, D*). These results confirm that BATF can alleviate lipid deposition in hepatocytes.

### Increased hepatic BATF alleviates HFD-induced hepatic steatosis in mice

We constructed an AAV8 virus overexpressing BATF to infect mouse liver. After infection, mice were fed normal chow or HFD. The weight and fat mass of the mice were measured weekly, and the mice were sacrificed 12 weeks after virus infection (*Figure 2A*). Overexpression of BATF increased hepatic BATF protein expression (*Figure 2B and C*), but there was no change in heart, muscle, fat, or spleen (*Figure 2D and E*). In addition, In the case of no difference in feed intake among the four groups (*Figure 2—figure supplement 1A*), HFD increased body weight and adiposity in mice, while BATF significantly alleviated weight gain and fat accumulation (*Figure 2F and G*). Mouse livers showed that HFD made them yellow and covered with fat-like particles, but BATF significantly alleviated these changes (*Figure 2H*). Liver hematoxylin and eosin (HE) staining of mice in the HFD group confirmed that BATF reduced the number of unstained vacuoles (lipid droplets) (*Figure 2I*). To verify the effect of BATF on hepatic lipid metabolism, oil red O staining, TG content and Glycerin level were examined and confirmed that BATF significantly alleviated HFD-induced lipid deposition (*Figure 2J–M*). BATF had no effect on the liver total cholesterol (TC) content of mice (*Figure 2—figure supplement 1B*). BATF did not have a significant impact on glucose metabolism in mice, including blood glucose levels (*Figure 2—figure supplement 1C*), Glucose Oxidase activity (*Figure 2—figure supplement 1D*) and glucose tolerance (*Figure 2— figure supplement 1E*, F).

We conducted further studies to explore the mechanism by which BATF regulated hepatic lipid metabolism. BATF reduced alanine aminotransferase (ALT) and aspartate aminotransferase (AST) activities (*Figure 3A and B*). It confirmed that BATF alleviated HFD-induced liver injury in mice. Detection of mRNA expression of lipid metabolism genes revealed that BATF did not affect lipid-synthesis-related genes (*Figure 3C*), but promoted expression of various lipid hydrolysis genes, including *AMPKα1, Aco, Acox1, Bcl2, Cpt1*, etc (*Figure 3D*). Fatty acid β-oxidation is a main route of fat metabolism, in which short-chain acyl coenzyme A dehydrogenase (SCAD) plays a key role. The result confirmed that BATF greatly promotes SCAD activity (*Figure 3E*). Fatty acid oxidation mainly takes place in mitochondria, so studies on the effect of BATF on aerobic oxidation were carried out. BATF elevated the level of ATP (*Figure 3F*). An oxygen consumption rate test was performed to explore the effect of BATF on mitochondrial oxygen consumption rate (*Figure 3G*), and indicated that BATF increased basal respiration, maximal respiration and ATP production in HepG2 cells (*Figure 3H*). These results suggest that BATF mitigates the effects of HFD on hepatic lipid deposition by promoting energy metabolism.

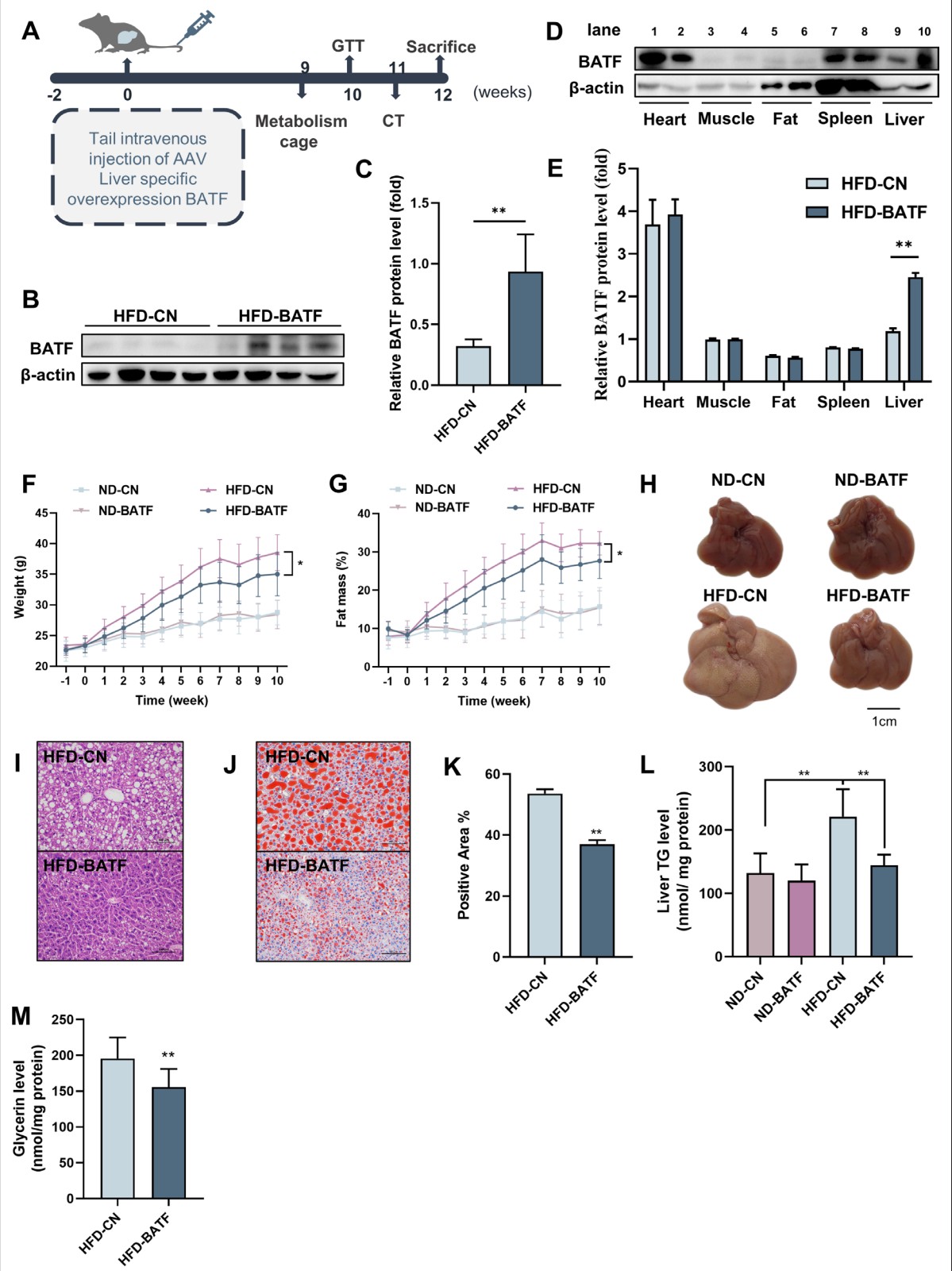

**Figure 2.** Effects of BATF on hepatic fat deposition in mice. (**A**) Experimental designs illustration of mice. (**B**) Expression of BATF protein in liver (n=4). (**C**) Densitometric quantification of the blotting. (**D**) Expression of BATF protein in various tissues of HFD-CN mice HBAAV/8-ZsGreen, WB in lane1, 3, 5, 7, 9 and HFD-BATF mice HBAAV2/8-CMV-m-BATF-3×flag-ZsGreen, WB in lane 2, 4, 6, 8, 10. (**E**) Densitometric quantification of the blotting. (**F**) Mice

*Figure 2 continued on next page*

*Figure 2 continued*

bodyweight (n=8–10). (**G**) Mice fat ratio (n=8–10). (**H**) Mice liver. Bar, 1 cm. (**I**) HE staining of mice liver sections. Bar, 100 µm. (**J**) (**K**) Oil red O staining of mice liver sections and quantitative analysis. Bar, 100 µm (n=3). (**L**) Liver triglyceride levels (n=8–10). (**M**) Liver total Glycerin levels (n=8–10).

The online version of this article includes the following source data and figure supplement(s) for figure 2:

**Source data 1.** Effects of BATF on hepatic fat deposition in mice.

**Figure supplement 1.** The effect of BATF on metabolic indicators in mice.

**Figure supplement 1—source data 1.** The effect of BATF on metabolic indicators in mice.

## BATF alleviates HFD-induced adipocyte hypertrophy in mice

BATF alleviated HFD-induced increase in adiposity (*Figure 2G*). After anesthetizing the mice, we used micro-computed tomography (CT) to scan the whole body and image the adipose tissue (*Figure 4A*). The red lightly shaded parts represent the adipose tissue of the mice, and the results indicated that BATF alleviated overall fat accumulation. The epididymal fat and subcutaneous fat of mice were separated. The results showed that BATF reduced the epididymal and subcutaneous fat (*Figure 4B–E*). HE staining of mouse adipose tissue allowed microscopic observation of adipocytes, and the diameter and area of the adipocytes indicated that BATF alleviated adipocyte hypertrophy (*Figure 4F–K*). BATF was overexpressed only in the liver but not in adipose tissue. To investigate the effect of BATF on lipid accumulation in adipocytes, we overexpressed BATF in 3T3-L1 cells and found that it had no effect on accumulation of TG (*Figure 4L*). These results confirm that BATF does not directly act upon adipocytes to reduce fat accumulation. Therefore, we speculated that BATF regulates fat accumulation in adipose tissue by influencing secretion of substances from the liver. To test this conjecture, we cultured differentiated 3T3-L1 cells with HepG2-cultured cell culture medium (overexpressing BATF) and assayed their TG content. The HepG2-cultured cell culture medium (overexpressing BATF) alleviated TG accumulation in 3T3-L1 cells (*Figure 4M*). Interleukin (IL)–27 has an inhibitory effect on fat accumulation. We confirmed that HFD decreased IL-27 expression, while BATF enhanced IL-27 expression (*Figure 4N*). This suggests that BATF mitigates the expansion of adipose tissue by promoting IL-27 secretion.

## BATF alleviates hepatocyte lipid accumulation by inhibiting programmed cell death protein 1

Several studies have confirmed that there is a regulatory relationship between BATF and programmed cell death protein (PD)1 (*Liu et al., 2019*; *Man et al., 2017* ; *Wu et al., 2019*). HFD increased PD1, while BATF inhibited PD1 expression in mouse liver (*Figure 5A*). These results were confirmed in HepG2 cells (*Figure 5B*). To determine the role of PD1 in lipid accumulation in hepatocytes, we overexpressed PD1 in HepG2 cells and assayed the lipid content. Both oil red O staining and TG content confirmed that PD1 promoted lipid deposition in hepatocytes (*Figure 5C and D*). PD1 antibody inhibited the role of PD1 and successfully alleviated TG accumulation in hepatocytes, but BATF overexpression plus PD1 antibody did not further alleviate lipid deposition (*Figure 5E*), suggesting that BATF regulates lipid deposition through PD1. To confirm this hypothesis, we overexpressed BATF and PD1. Overexpression of BATF decreased TG, while the TG-lowering effect of BATF disappeared when PD1 was overexpressed (*Figure 5F*). To determine the regulatory role of BATF on PD1, we constructed a luciferase vector containing 2000 bp upstream of the transcription start site of PD1. BATF decreased the transcriptional activity of PD1 promoter, confirming the role of BATF in regulating PD1 expression (*Figure 5G*). These results suggest that BATF regulates lipid deposition through inhibition of PD1. In conclusion, HFD leads to an increase in the level of PD1 transcription in the liver. BATF promotes lipolysis and energy consumption in hepatocytes by regulating PD1 transcription, thereby reducing HFD-induced liver lipid deposition and alleviating NAFLD (*Figure 5H*).

## PD1 antibody alleviates HFD-induced obesity and liver steatosis in mice

PD1 plays an important role in tumorigenesis as a tumor immune escape target, and PD1 antibodies as antitumor drugs can inhibit PD1 and mitigate tumor growth and cancer development (*Ribas and Wolchok, 2018*). Our study confirmed that PD1 promoted lipid deposition in hepatocytes (*Figure 5C and D*), and in vivo experiments were performed to verify the role of PD1 on hepatic steatosis in mice

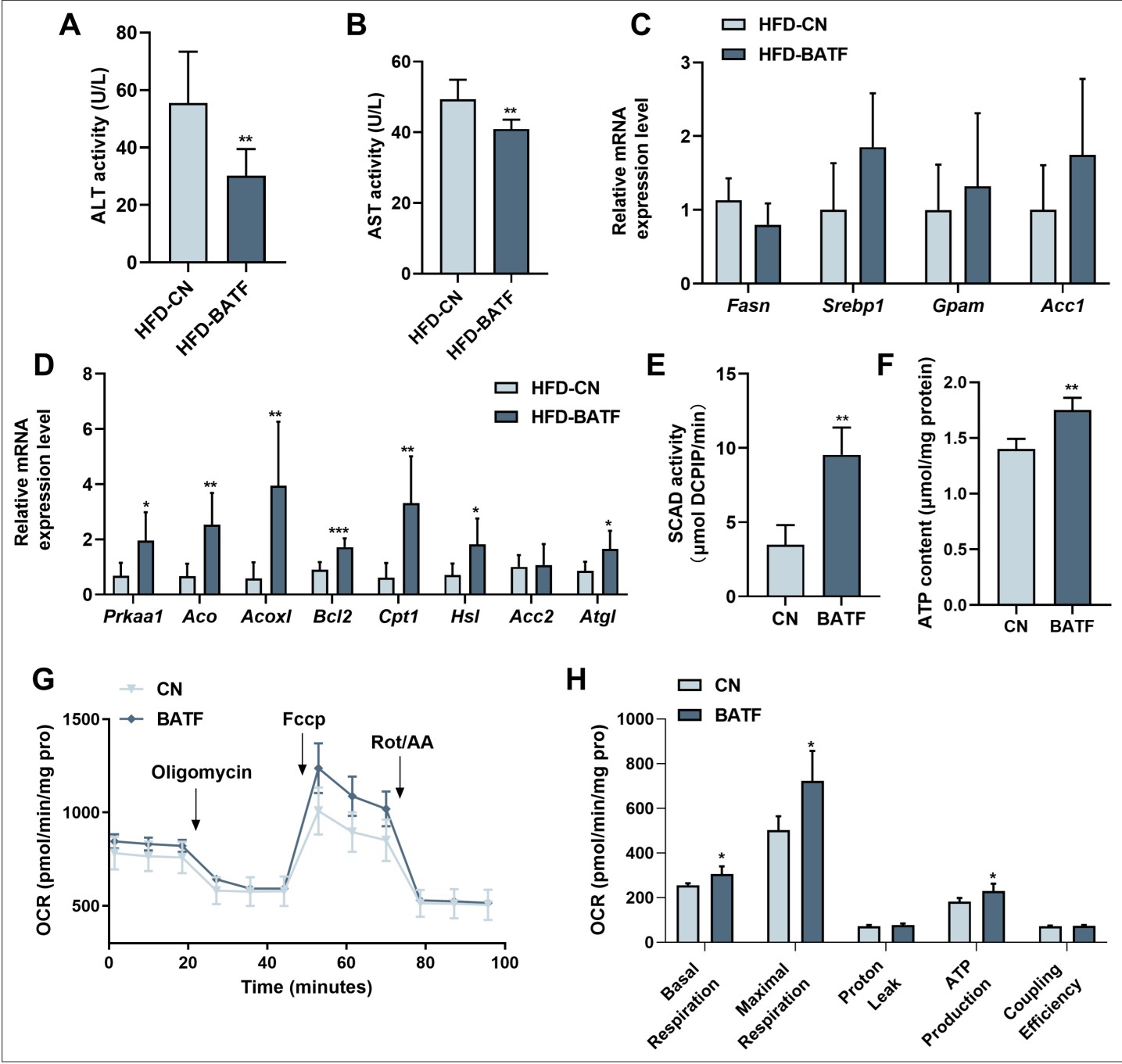

**Figure 3.** BATF boosts lipid breakdown and energy metabolism in mice livers. (**A**) ALT activity in mice liver (n=8). (**B**) AST activity in mice liver (n=7). (**C**) The *Fasn*, *Srebp1*, *Gpam*, *Acc1* mRNA expression level in mice liver (n=6–8). (**D**) The *AMPKα1*, *Aco*, *Acox1*, *Bcl2*, *Cpt1*, *Hsl*, *Acc2*, *Atgl* mRNA expression level in mice liver (n=6–7). (**E**) SCAD activity in HepG2 cells with OA/PA treatment (n=4). (**F**) ATP content in HepG2 cells with OA/PA treatment (n=4). (**G**) Oxygen consumption rate (OCR). (**H**) Basal respiration, maximal respiration, proton leak and coupling efficiency. The data are expressed as mean ± SD. *p<0.05, **p<0.01.

The online version of this article includes the following source data for figure 3:

**Source data 1.** BATF boosts lipid breakdown and energy metabolism in mice livers.

fed an HFD. PD1 antibody alleviated HFD-induced weight gain in mice after intraperitoneal injection of PD1 antibody (200 μg; *Figure 6A*). The mice injected with PD1 antibody were smaller than the control mice (*Figure 6B*). The results of fat mass confirmed that PD1 antibody alleviated fat accumulation in mice (*Figure 6C*). The epididymal and subcutaneous fat of mice were reduced by PD1 antibody

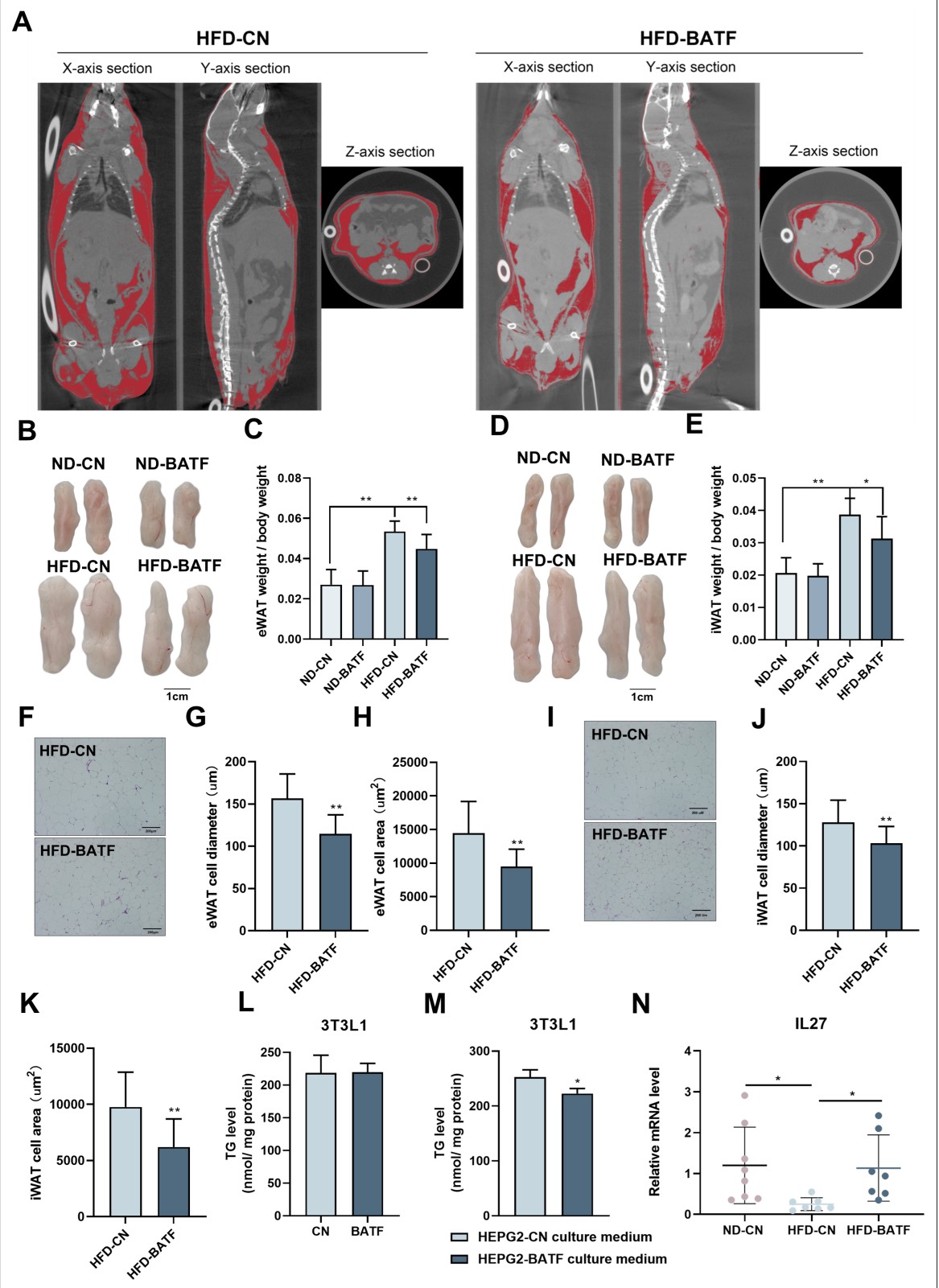

**Figure 4.** BATF alleviates HFD-induced adipocyte hypertrophy in mice. (**A**) CT images of fat axial view. (**B**) eWAT of mice. Bar, 1 cm. (**C**) eWAT weight / bodyweight (n=9–10). (**D**) iWAT of mice. Bar, 1 cm. (**E**) eWAT weight / bodyweight (n=9–10). (**F**) HE staining of eWAT, (**G**) adipocyte diameter and (**H**) cell area. Bar, 200 μm. (**I**) HE staining of iWAT, (**J**) adipocyte diameter and (**K**) cell area. Bar, 200 μm. (**L**) Triglyceride content of undifferentiated 3T3L1 cells

*Figure 4 continued on next page*

*Figure 4 continued*

(n=5). (**M**) Triglyceride content of differentiated 3T3L1 cells (n=3–4). (**N**) The mRNA expression of IL27 in liver tissues (n=5). The data are expressed as mean ± SD. *p<0.05, **p<0.01.

The online version of this article includes the following source data for figure 4:

**Source data 1.** BATF alleviates HFD-induced adipocyte hypertrophy in mice.

(*Figure 6D–G*). Mouse livers showed that HFD gave them an appearance of hepatic steatosis, which was significantly alleviated by PD1 antibody (*Figure 6H*). The liver was stained with Oil Red O and assayed for TG content, which showed that PD1 antibody reduced accumulation of lipid droplets (*Figure 6I*) in the liver and alleviated lipid deposition (*Figure 6J*). We extracted RNA from mouse livers and examined expression of lipolysis-related genes. PD1 antibody treatment was consistent with over-expression of BATF, and both promoted expression of lipolysis-related genes (*Figure 6K*). Similarly, PD1 antibody treatment promoted SCAD activity in mouse liver, suggesting fatty acid β-oxidation was enhanced (*Figure 6L*).

## Discussion

In the present study, we found that HFD elevated BATF expression in mouse liver. We also found that inhibition of hepatocyte BATF did not regulate lipid deposition, while overexpression of BATF decreased TG accumulation. Studies on hepatocytes confirmed that BATF regulated hepatocyte lipid deposition. BATF also alleviated HFD-induced obesity in mice, not by directly acting on adipose tissue but by influencing IL-27, which acts as a regulator of fat deposition (*Wang et al., 2021*). The regulation of hepatic lipid metabolism by BATF has not previously been investigated. Here, we found that BATF mitigated HFD-induced increase in PD1 expression. We revealed that PD1 promoted hepatocyte lipid deposition. Through luciferase assays, we found that BATF regulated lipid deposition by inhibiting PD1 expression. Experiments in mice confirmed that PD1 inhibition alleviated HFD-induced obesity and hepatic steatosis.

The adaptive immune system has evolved to eliminate virtually any threat from the organism (*Sharpe and Pauken, 2018*). PD1 is expressed during T-cell activation and through T-cell receptor counter-acting positive signaling by engaging its ligands PDL1 and/or PDL2 (*Freeman et al., 2000*). PD1 regulates the immune system and promotes self-tolerance by downregulating the immune response to human cells and by suppressing T-cell inflammatory activity (*Lei et al., 2020*). The development of checkpoint blockade inhibitors has revolutionized cancer immunotherapy (*Sharma and Allison, 2015*) and has led to durable survival outcomes in some patients with metastatic disease (*Pulendran and Davis, 2020*). Previous studies on PD1 have focused on the immune system, and we found that HFD increased its expression in the liver of mice (*Figure 5A*). This is consistent with previous studies (*Wang et al., 2019*) suggesting that PD1 is equally important in the liver. Several studies have demonstrated that PD1 and BATF regulate each other to modulate immune system flexibility (*Quigley et al., 2010*; *Ji et al., 2020*). We hypothesized that BATF and PD1 co-regulate the hepatic immune system to affect lipid deposition. Our results confirm our conjecture that BATF reduces fat accumulation by inhibiting PD1, confirming that PD1 is downstream of the BATF signaling pathway (*Figure 5F and G*).

Cancer and autoimmune disease are closely related, and many therapeutic antibodies are widely used in clinics for the treatment of both diseases. Moreover, immune checkpoint blockade using the anti-PD1/PD-L1/CTLA4 antibody has been shown to improve prognosis of patients with refractory solid tumors *Yasunaga, 2020*. In this study, we used PD1 antibodies to treat HFD-induced obesity and NAFLD in mice. Surprisingly, PD1 antibody has an excellent therapeutic effect on obesity and NAFLD. PD1 antibodies reduced adiposity and hepatic lipid deposition (*Figure 6*), which is consistent with overexpression of BATF. There have been no approved drugs for NAFLD treatment. In this study, the effect of PD1 antibodies on NAFLD treatment was encouraging. Our findings have somewhat pioneered the cognition of BATF, PD1 and its expansion with NAFLD therapy.

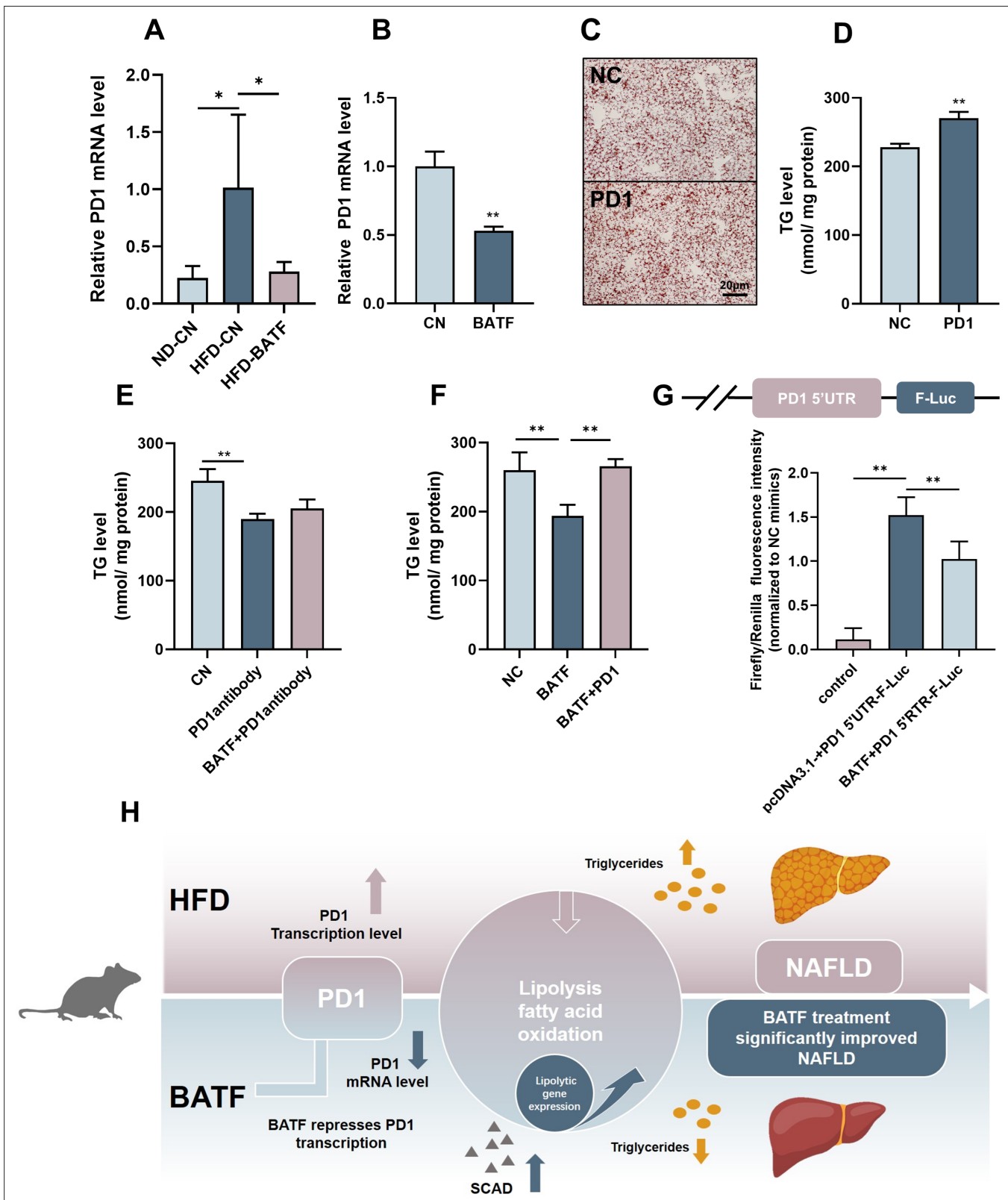

**Figure 5.** BATF alleviates hepatocyte lipid accumulation by inhibiting PD1. (**A**) The mRNA expression of PD1 in liver tissues (n=3). (**B**) The mRNA expression of PD1 in HepG2 cells (n=3). (**C**) Oil red O staining of HepG2 cells, Bar, 20 μm, (n=3). (**D**), (**E**), (**F**) Triglyceride content with OA/PA (n≥3). (**G**) Dual luciferase assay on Hepa1-6 cells cotransfected with firefly luciferase constructs containing the PD1 promoter, Renilla luciferase vector pRL-

*Figure 5 continued on next page*

*Figure 5 continued*

TK and pcDNA3.1(-) or pcDNA3.1(-)-BATF, (n≥3). (**H**) The Mechanism diagram of BATF alleviates hepatocyte lipid accumulation by PD1. The data are expressed as mean ± SD. *p<0.05, **p<0.01.

The online version of this article includes the following source data for figure 5:

**Source data 1.** BATF alleviates hepatocyte lipid accumulation by inhibiting PD1.

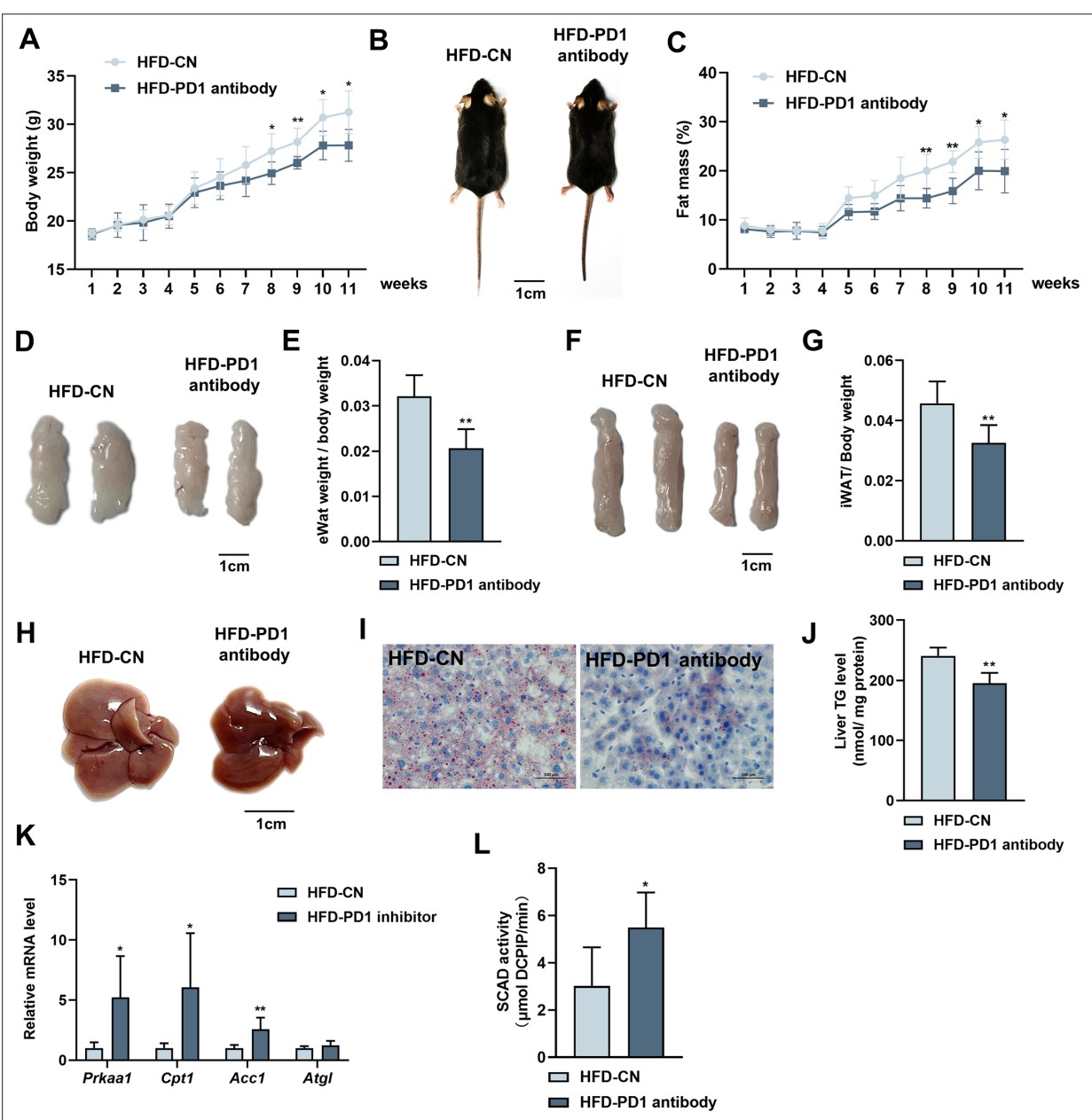

**Figure 6.** Effects of PD1 antibody on liver lipid metabolism in HFD mice. (**A**) Mice bodyweight (n≥5). (**B**) The mice were injected with IgG or PD1 antibody under HFD (n≥5). (**C**) Mice fat ratio (n≥5). (**D**) eWAT of mice. Bar, 1 cm. (**E**) eWAT weight / bodyweight (n≥5). (**F**) iWAT of mice. Bar, 1 cm. (**G**) eWAT weight / bodyweight (n≥5). (**H**) The liver of mice. (**I**) HE staining of mice liver sections. Bar, 100 μm. (**J**) Liver triglyceride (TG) levels. (**K**) The *AMPKα1*, *Cpt1*, *Acca*, *Atgl* mRNA expression level in mice liver (n≥5). (**L**) SCAD activity in liver of mice (n=5). The data are expressed as mean ± SD. *p<0.05, **p<0.01.

The online version of this article includes the following source data for figure 6:

**Source data 1.** Effects of PD1 antibody on liver lipid metabolism in HFD mice.

# Materials and methods

**Key resources table**

| Reagent type (species) or resource | Designation | Source or reference | Identifiers | Additional information |
|---|---|---|---|---|
| Gene (*Mice*) | BATF | GenBank | NM_016767.2 | |
| Strain, strain background (*Mus musculus*) | C57BL/6 J (*Wild type*) | Guangxi Medical University | | male |
| Cell line (Human) | HepG2 (hepatocellular carcinoma, youth) | ATCC | ATCC number: HB-8065 | |
| Cell line (Human) | L02 (Normal, fetal liver) | YUCHI Biology | | |
| Cell line (*Mus musculus*) | Hepa1-6 (Hepatoma) | ATCC | ATCC number: CRL-1830 | |
| Cell line (*Mus musculus*) | AML12 (Normal, 3-month-old) | ATCC | ATCC number: CRL-2254 | |
| Cell line (Human) | HEK293T (Normal, Kidney) | ATCC | ATCC number: ACS-4500 | |
| Cell line (*Mus musculus*) | 3T3L1 (Embryo) | ATCC | ATCC number: CL-173 | |
| Transfected construct (*Mus musculus*) | HBAAV/8-ZsGreen | packaged by HANBIO | | |
| Transfected construct (*Mus musculus*) | HBAAV2/8-CMV-m-BATF-3×flag-ZsGreen | packaged by HANBIO | | Adeno-associated virus construct to transfect and express the BATF. |
| Biological sample (*Mus musculus*) | primary hepatocytes | This paper | Materials and methods: Primary hepatocyte isolation and AAV infection | Freshly isolated from C57BL/6 J |
| Antibody | BATF (D7C5) (Rabbit monoclonal) | Cell signaling technology | Cat# 8638 | WB (1:1000) |
| Antibody | Anti-beta Actin (Rabbit polyclonal) | Servicebio | Cat# GB11001 | WB (1:1000) |
| Antibody | Peroxidase AffiniPure Goat Anti-Rabbit IgG (H+L) (Rabbit Polyclonal) | Jackson ImmunoResearch | Cat# 111-035-003 | WB (1:2000) |
| Recombinant DNA reagent | pcDNA3.1(-)-Mice BATF | This paper | Cell culture and transfection | |
| Recombinant DNA reagent | pcDNA3.1(-)-Mice PD1 | This paper | Cell culture and transfection | |
| Recombinant DNA reagent | pMSCV-PIG-BATF | This paper | Retroviral transduction | |
| Recombinant DNA reagent | pUMVC | Add gene | Plasmid #8449 | |
| Recombinant DNA reagent | pCMV-VSV-G | Add gene | Plasmid # 8454 | |
| Recombinant DNA reagent | pGL3-PD1 vector | This paper | Luciferase assay for promoter activity analysis | |
| Recombinant DNA reagent | pRL-TK | Beyotime Biotechnology | D2760 | |
| Sequence-based reagent | siRNA: BATF RNAi | synthesized from Sangon Biotech | | |
| Commercial assay or kit | Triglyceride assay kit | Nanjing Jiancheng Bioengineering Institute | Cat# A110-1-1 | |
| Commercial assay or kit | Total cholesterol | Nanjing Jiancheng Bioengineering Institute | Cat# A111-1-1 | |
| Commercial assay or kit | Tissue SCAD assay kit | Gen Med | Cat# GMS 50119.2.1 | |

*Continued on next page*

*Continued*

| Reagent type (species) or resource | Designation | Source or reference | Identifiers | Additional information |
|---|---|---|---|---|
| Commercial assay or kit | Alanine transaminase (ALT) | Nanjing Jiancheng Bioengineering Institute | Cat# C009-2-1 | |
| Commercial assay or kit | Aspartate aminotransferase (AST) | Nanjing Jiancheng Bioengineering Institute | Cat# C010-2-1 | |
| Commercial assay or kit | Cellular ATP content | Beyotime Biotechnology | Cat# S0026 | |
| Commercial assay or kit | RNA Reverse Transcription System | Promega | Cat# A3500 | |
| Commercial assay or kit | ClonExpress II One Step Cloning Kit | Vazyme | Cat# C112-01/02 | |
| Commercial assay or kit | Dual-Lumi II Luciferase Reporter gene Assay kit | Beyotime Biotechnology | Cat# PG089S | |
| Commercial assay or kit | Cellular ATP content | Beyotime Biotechnology | Cat# S0026 | |
| Chemical compound, drug | High-fat-diet (HFD) | Trophic Animal Feed High-tech Co. Ltd | TP2330055M | standar: D12492 |
| Chemical compound, drug | Normal diet (ND) | Trophic Animal Feed High-tech Co. Ltd | TP2330055MC | standar: AIN93 |
| Chemical compound, drug | Oleic acid (OA) | Sigma-Aldrich | Cat# O1383 | |
| Chemical compound, drug | Palmitic acid (PA) | Sigma-Aldrich | Cat# P0500 | |
| Software, algorithm | Image J | National Institutes of Health | https://Imagej.nih.gov/ij/ | |
| Software, algorithm | Adobe Photoshop CS6 | Adobe | | https://www.adobe.com/cn |
| Software, algorithm | GraphPad Prism 8.0.1 | GraphPad | | https://www.graphpad.com/ |
| Other | InVivoPlus anti-mouse PD1 (CD279) (Mouse RMP1-14) | Bio X Cell | Cat# BP0146 | Intraperitoneal injection: 200 µg/mouse |
| Other | Rat IgG-Isotype Control (Rat monoclonal) | abcam | Cat# ab37361 | Intraperitoneal injection: 200 µg/mouse |

## Animals

All animal studies were approved by the Animal Ethics Committee of Guangxi University (GXU-2020–288). Four-week-old male C57BL/6 J mice were purchased from Guangxi Medical University (Nanning, China) and kept in individual cages on a 12 hr light/dark cycle with free access to water and food at room temperature. At week 10, mice were injected intravenously (tail vein) with 100 µl adeno-associated virus (AAV). AAV was HBAAV2/8-CMV-m-BATF-3×flag-ZsGreen and HBAAV/8-ZsGreen (packaged by HANBIO, Shanghai, China). The mice were fed with a normal diet (ND) (AIN93) or HFD (D12492) for 12 weeks (*Qi et al., 2020*). To study the effect of PD1 inhibitors on NALFD, the mice were fed with HFD and rat IgG (ab37361; Abcam) or anti-mouse PD1 (AB10949053, Bio X Cell) antibody was administered (intraperitoneal injection) at 200 µg/mouse twice weekly (*Zelenay et al., 2015*). Body weight and fat mass were recorded weekly, and body composition (fat) was determined by Body Composition Analyzer (QMR-23–060 H-I; Suzhou, China). Mice were anesthetized with 2.4% tribromoethanol (T48402, Sigma–Aldrich) to evaluate fat content, and micro-CT images were taken using a micro-CT scanner (SKYSCAN1278; Bruker, Belgium). All groups were randomized and blinded to form equal-sized groups of at least three (exact numbers are given in the legend).

## Human disease data

The human data of NAFLD were taken from the Gene Expression Omnibus, Accession Number GSE130970 (*Hoang et al., 2019*).

## Cell culture and transfection

HepG2 (HB-8065, ATCC), L02 (YUCHI Biology, China), Hepa1-6 (CRL-1830, ATCC), HEK293T (ACS-4500, ATCC), 3T3L1 (CL-173, ATCC) cells were cultured in DMEM medium (C11995500BT, Gibco)

supplemented with fetal bovine serum (04-400-1A, BI) and 1% penicillin–streptomycin (P1400, Solarbio, China). The AML12 cell (CRL-2254, ATCC) culture medium was DMEM/F12 (Gibco) containing 10% FBS and 1% penicillin–streptomycin. The cells were incubated at 37 °C and 5% carbon dioxide. For simulating HFD in vitro, 200 µM OA and 100 µM PA were conjugated to bovine serum albumin (BSA) as previously described (*Li et al., 2019*).

For transfection of plasmids or siRNA oligos, Hieff Trans Liposomal Transfection Reagent (40802ES02, YEASEN) was used. Plasmid constructs were created using the eukaryotic expression vector pcDNA3.1- (*Liang Huang et al., 2022*). Mice BATF (NM_016767.2), PD1 (NM_008798.3) CDS were cloned from mouse liver cDNA. RNAi sequence was synthesized from Sangon Biotech (Shanghai, China).

## Primary hepatocyte isolation and AAV infection

The abdominal cavity of 8-week-old male mice was opened after anesthesia with 2.4% tribromoethanol. The inferior vena cava was cannulated with a 23 G needle and perfused with 30 mL perfusion buffer (1×HBSS without $Ca^{2+}$, $Mg^{2+}$ supplemented with 0.5 mM EDTA and 25 nM HEPES). The liver was digested by 25 mL of 37°C digestion buffer (1×HBSS with $Ca^{2+}$, $Mg^{2+}$ supplemented with 1 µg/mL collagenase type II and 1 M HEPES). Following the perfusion, the liver was gently removed and massaged through 70 µm nylon mesh with 10 mL of 4°C precooled complete medium (DMEM supplemented with 1% FBS). Hepatocytes were isolated by centrifugation at 50 $g$, 4 °C, for 2 min. After discarding the supernatant, hepatocytes were resuspended in 10 mL complete medium containing 5 mL freshly prepared 90% Percoll solution (9 mL Percoll with 1 mL PBS). Cell debris was removed by centrifugation at 200 $g$, 4 °C, for 10 min. The supernatant was aspirated, resuspended and centrifuged at 50 $g$, 4°C for 2 min, after resuspension in William E medium supplemented 1% glutamine and 1% penicillin–streptomycin. Cell resuspension was transferred into a 0.1% gelatin-coated plate and incubated at 37 °C, 5% carbon dioxide (*Zhang et al., 2021b*). Cells were maintained for 24 hr in medium. For in vitro experiments with AAV, 1 µL f purified HBAAV2/8-CMV-m-BATF-3×flag-ZsGreen or HBAAV/8-ZsGreen was added to cultured hepatocytes to achieve high BATF overexpression.

## Glucose tolerance test

Glucose tolerance test (GTT) was performed as previously described (*Zhang et al., 2022*). Mice were fasted for 16 hr and injected with D-glucose at 2 g/kg intraperitoneally. The blood glucose level was measured at 0,15, 30, 60, and 120 min after glucose injection.

## Western blotting

Tissues were lysed with 1 mL RIPA lysis buffer and supplemented 1% protease inhibitor cocktail (Solarbio) for 2–5 min with steel balls in a tissue lyser (Qiagen, Germany). Tissue lysates were incubated with the buffer at 4 °C overnight (cell samples were incubated in lysis buffer for 30 min at 4 °C). The lysate was centrifuged at 12,000 rpm for 10 min, the supernatant was collected, and the protein concentration was determined using the BCA method (P0011, Beyotime Biotechnology). The supernatant was mixed with loading buffer and boiled at 100 °C for 10 min. Samples were loaded and subjected to 10% SDS-PAGE. After transfer to PVDF membranes, blots were blocked with 5% skimmed milk powder and incubated with primary antibody overnight at 4 °C. The primary antibodies included a BATF antibody (8638 S, Cell Signaling Technology, USA), anti-β-actin (GB11001, Servicebio, China), and β-tublin (15115, Cell Signaling Technology). Secondary antibody was goat anti-rabbit (Jackson, 111-035-003), images were acquired using the BIO-RAD protein gel imaging system (Universal hood II), and band intensity was calculated using Image J (*Zhang et al., 2021c*).

## TGs, TC, and glycerin detection

Cells or tissues were prepared according to the instructions of the A110-1-1 Triglyceride Assay Kit (*Zhang et al., 2021a*), A111-1-1 Total Cholesterol Assay Kit (Nanjing Jiancheng Bioengineering Institute, China), and Tissue Cell Glycerin Test kit (E1024, Pulilai, China). Liver tissues or cells were lysed by RIPA lysis buffer. Samples were centrifuged at 12,000 rpm for 10 min, and the supernatants were collected. Supernatants (10 µL) were added to a 96-well plate and 190 µL working solution was added to each well. The plate was incubated at 37 °C for 10 min. The microplate reader (Tecan M200 PRO) recorded the contents of TG, TC and glycerin, then normalized using the protein concentration.

## Enzyme activity assay

SCAD enzyme activity in cultured cells or liver tissues was detected using the SCAD Assay Kit (Gen Med, Plymouth, MN, USA). The liver tissue activities of ALT and AST were measured using the C009-2-1 ALT Reagent Kit and C010-2-1 AST Reagent Kit (Nanjing Jiancheng Bioengineering Institute, China). The liver tissue activities of glucose oxidase was measured using the BC0695 Glucose Oxidase(GOD) Activity Assay Kit (Servicebio, China).

## ATP measurements

Cellular ATP content was measured using a luciferin/luciferase-based kit (S0026, Beyotime Biotechnology, China).

## HE and Oil Red O staining

HE and Oil Red O staining was performed as previously described (*Luo et al., 2021*). Livers, epididymal fat and subcutaneous fat of mice were dissected and fixed in 4% paraformaldehyde overnight at 4 °C. Samples were sent to Wuhan Service Technology Co. Ltd. (Wuhan, China) for paraffin embedding, sectioning, and staining. Images were collected by light microscope and Image J was used to examine the cell diameter and the area.

## Oxygen consumption rate

Oxygen consumption rate was measured by extracellular flux (XF24) analyzer (Seahorse Bioscience). Cells were seeded at $2 \times 10^4$ per well in XF24 plates. The probe plate was hydrated with the calibration solution (pH 7.4) and a 2 mM glutamine assay solution (pH 7.35±0.05) was prepared, and incubated for 16–20 hr at 37 °C and 5% $CO_2$. Culture medium was replaced by 2% FBS medium and incubated without $CO_2$ for 1 hr before the completion of probe cartridge calibration. Basal oxygen consumption rate was measured after injection of oligomycin (1 μM), carbonyl cyanide p-trifluoro-methoxyphenyl hydrazone (FCCP 1 μM), and rotenone/antimycin A (ROT/AA), (0.5 μM; *Liu et al., 2021*).

## RNA extraction and real-time quantitative polymerase chain reaction

Total RNA was extracted by TRIzol Reagent (Life Technologies, USA). Reverse transcription of RNA was performed with the Reverse Transcription System (A3500, Promega), and formed cDNA from 1 μg total RNA. 2×RealStar Green Fast Mixture was used for real-time quantitative polymerase chain reaction (PCR) (GenStar, Beijing, China). The relevant primer sequences (5′–3′) were as follows: *ACTB* forward: CACCATTGGCAATGAGCGGTTC, reverse: AGGTCTTTGCGGATGTCCACGT. Human BATF forward: GATGTGAGAAGAGTTCAGAGGAG, reverse: GTTTCTCCAGGTCTTCGCTCTC. Mouse *Batf* forward: ATGCCTCACAGCTCCGACAGC, reverse: TCAGGGCTGGAAGCGTGGC. Mouse PD-1 forward: CGGTTTCAAGGCATGGTCATTGG, reverse: TCAGAGTGTCGTCCTTGCTTCC; Mouse *Fasn* forward: CACAGTGCTCAAAGGACATGCC, reverse: CACCAGGTGTAGTGCCTTCCTC. Mouse *Srebp1* forward: CGACTACATCCGCTTCTTGCAG, reverse: CCTCCATAGACACATCTGTGCC. Mouse *Gpam* forward: GCAAGCACTGTTACCAGCGATC, reverse: TGCAATCAGCCTTCGTCGGAAG. Mouse *Acc1* forward: GTTCTGTTGGACAACGCCTTCAC, reverse: GGAGTCACAGAAGCAGCCCATT. Mouse *Prkaa1* forward: GGTGTACGGAAGGCAAAATGGC, reverse: CAGGATTCTTCCTTCGTACACGC. Mouse Aco forward: TCACAGCAGTGGGGATTCCAA, reverse: TCTGCAGCATCATAACAGTGTTCTC. Mouse *Acoxl* forward: TAACTTCCTCACTCGAAGCCA, reverse: AGTTCCATGACCCATCTCTGTC. Mouse *Bcl2* forward: ACTTCCACTACAGGACAGAC, reverse: TCTAAGGTGACTCGATATGG. Mouse *Cpt1* forward: GCACTGCAGCTCGCACATTACAA, reverse: CGTTGACATCCGTAAAGACC. Mouse *β-actin* forward: AACAGTCCGCCTAGAAGCAC, reverse: CTCAGACAGTACCTCCTTCAGGAAA. Mouse *Hsl* forward: GGAGCACTACAAACGCAACGA, reverse: TCGGCCACCGGTAAAGAG. Mouse *Acc2* forward: AGAAGCGAGCACTGCAAGGTTG, reverse: GGAAGATGGACTCCACCTGGTT. Mouse *Atgl* forward: GGAACCAAAGGACCTGATGACC, reverse: ACATCAGGCAGCCACTCCAACA.

## Retroviral transduction

Retroviral vectors pMSCV-PIG and pMSCV-PIG-BATF were used to infect 3T3L1 cells. The pMSCV-PIG-BATF viral vector was constructed, and the coding sequence of mouse BATF was amplified from liver cDNA by PCR and cloned into the *Eco*RI site using ClonExpress II One Step Cloning Kit (C112-01/02, Vazyme, China). To obtain pseudotyped virus with the ability to infect, pMSCV-PIG-BATF,

pMSCV-PIG-GFP, and retroviral packaging plasmids (pUMVC and pCMV-VSV-G) were cotransfected into subconfluent HEK293T cells. The virus stocks were collected 2 and 3 days after transfection, centrifuged at 5000 $g$ for 10 min, subpackaged and frozen at −80 °C. 3T3L1 cells with around30% confluence were treated with virus stock containing 8 µg/mL polybrene for 4 hr, washed, and cultured in fresh medium.

## Luciferase assay for promoter activity analysis

Firefly luciferase constructs were cloned using the pGL3-Basic vector as a backbone. Target 5′ UTRs (PD1 was amplified by primers, forward: AGCATGAGCCCTGAGGATTG reverse: CAGTGTCGCCTT CAGTAGCA) were cloned upstream of Firefly luciferase in the pgl3. Renilla luciferase reporter vector pRL-TK was gifted by State Key Laboratory for Conservation and Utilization of Subtropical Agro-Bioresources. Hepa1-6 cells were inoculated into medium in 24-well culture plates and cultured overnight. Cell transfection and cotransfection were conducted when the cell density reached 80%. Dual Luciferase Reporter Assay was performed 48 hr post-transfection using the Dual-Lumi II Luciferase Reporter gene Assay kit (PG089S). Renilla luciferase was used to normalize transfection efficiency and luciferase activity.

## Statistical analysis

All experiments were repeated at least in triplicate, and statistical analysis was done using independent values. All data are presented as mean ± SD. Statistical analysis was performed using Student's unpaired two-tailed $t$ test (for two groups) or analysis of variance (ANOVA) (for multiple groups). One-way ANOVA with Tukey's test was conducted for comparisons. The differences were considered significant at $p < 0.05$ (*). Values with different letters were significantly different.

## Acknowledgements

This work was supported by National Natural Science Foundation of China (32272952); Guangxi Science Foundation for Distinguished Young Scholars (2020GXNSFFA297008); Guangxi Natural Science Foundation (2019GXNSFDA245029); Guangxi Academy of Medical Sciences high-level Talents Foundation (YKY-GCRC-202302).

## Additional information

### Funding

| Funder | Grant reference number | Author |
| --- | --- | --- |
| National Natural Science Foundation of China | 32272952 | Lei Zhou |
| Guangxi Science Foundation for Distinguished Young Scholars | 2020GXNSFFA297008 | Lei Zhou |
| Guangxi Natural Science Foundation | 2019GXNSFDA245029 | Lei Zhou |
| Guangxi Academy of Medical Sciences high-level Talents Foundation | YKY-GCRC-202302 | Lei Zhou |

The funders had no role in study design, data collection and interpretation, or the decision to submit the work for publication.

### Author contributions

Zhiwang Zhang, Qichao Liao, Conceptualization, Data curation, Validation, Visualization, Methodology, Writing – original draft, Writing – review and editing; Tingli Pan, Songtao Su, Shi Liu, Menglong Hou, Yixing Li, Turtushikh Damba, Yunxiao Liang, Investigation; Lin Yu, Software; Zupeng Luo, Visualization;

Lei Zhou, Conceptualization, Resources, Data curation, Formal analysis, Funding acquisition, Validation, Investigation, Methodology, Writing – review and editing

**Author ORCIDs**
Lei Zhou ⓘ http://orcid.org/0000-0002-7145-4953

**Ethics**
This study was performed in strict accordance with the recommendations in the Guide for the Care and Use of Laboratory Animals of the National Institutes of Health. All animal studies were approved by the Animal Ethics Committee of Guangxi University (GXU-2020-288). All surgery was performed under sodium pentobarbital anesthesia, and every effort was made to minimize suffering.

Reviewer #1 (Public Review): https://doi.org/10.7554/eLife.88521.3.sa1
Reviewer #2 (Public Review): https://doi.org/10.7554/eLife.88521.3.sa2
Author Response https://doi.org/10.7554/eLife.88521.3.sa3

## Additional files

**Supplementary files**
• MDAR checklist

**Data availability**
All data generated or analysed during this study are included in the manuscript and supporting file. Source Data files have been provided for Figures 1-6 and Figure 1-2 Figure supplement.

The following previously published dataset was used:

| Author(s) | Year | Dataset title | Dataset URL | Database and Identifier |
|---|---|---|---|---|
| Hoang SA, Oseini A, Feaver RE, Cole B, Asgharpour A, Vincent R, Siddiqui M, Lawson MJ, Day NC, Taylor JM, Wamhoff BR, Mirshahi F, Contos MJ, Idowu M, Sanyal AJ | 2019 | Gene expression predicts histological severity and reveals distinct molecular profiles of nonalcoholic fatty liver disease | https://www.ncbi.nlm.nih.gov/geo/query/acc.cgi?acc=GSE130970 | NCBI Gene Expression Omnibus, GSE130970 |

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
