## [Editor Report · eLife assessment]

This **valuable** study presents reports on the role of the transcription factor BATF and its target PD1 in lipid metabolism including a model of nonalcoholic fatty liver disease (NAFLD). Overall, the evidence supporting the conclusions is **convincing**. The work will be of interest to medical biologists working on NAFLD.

---

## [Referee Report · Reviewer #1 (Public Review)]

The authors investigated the function of BATF in hepatic lipid metabolism. They found BATF alleviated high-fat diet (HFD)-induced hepatic steatosis. In addition, BATF could inhibit programmed cell death protein (PD)1 expression induced by HFD. By using over expression and transcriptional activity analysis, this study confirmed that BATF regulates fat accumulation by inhibiting PD1 expression and promoting energy metabolism. Then, they found PD1 antibodies alleviated hepatic lipid deposition. These data identified the regulatory role of BATF in hepatic lipid metabolism and that PD1 is a target for alleviation of NAFLD. The conclusions of this manuscript are supported by the data.

---

## [Referee Report · Reviewer #2 (Public Review)]

In this manuscript, authors firstly investigated the role of a transcriptional factor BATF in hepatic lipid metabolism both in vivo and in vitro. By using a AAV transfection to overexpress BATF in liver, the mice with overexpression of BATF resisted the high fat diets induced obesity and attenuated the hepatic steatosis. Mechanically, the PD1 mediated its effect on lipid accumulation in hepatocyte and IL-27 mediated its effect on adiposity reduction in vivo.

Strengths:

1. This work found the transcription factor BATF was positive to reduce hepatic lipid accumulation and offered a potential target to treat NAFLD.

2. PD1 antibody is always used to treat cancer, authors here have developed its new function in metabolic disease. PD1 antibody could help mice to combat obesity and hepatic steatosis induced by high fat diets.

3. Overexpression of BATF in the liver not only decreased the lipid accumulation in the liver but also reduced the fat mass. IL-27 secretion in the liver was enhanced to affect the adipose tissue. The cross talk in liver and adipose tissue was also validated in this paper.

---

## [Author Response]

The following is the authors’ response to the original reviews.

**Reviewer #1 (Public Review):**

Thank you very much for your advices and comments. We took your suggestion into consideration and decided to modify it as you suggested. We will add more data and analysis on this topic in the article to make the exposition fuller.

1. There are different cells in liver tissue, in which BATF protein is expressed most.

Based on the analysis of single-cell public data (GEO accession: GSE129516), BATF is expressed in every cell cluster in the liver, with the highest expression in T cells and the least in cholangiocytes (Author response image 1).

**Author response image 1. sa3fig1:** The expression level of BATF in clumster of cells in the liver.

1. The statistical data should be provided to support the liver specific over-expression of BATF.

The results of WB in figure2 (C & E) have been quantified and relevant content has been corrected.

1. For in vivo study, food intake is key data to exclude the change of energy intake.

Feed intake related result plots have been added to figureS2A.

1. For Fig.6 Since PD1 are also highly expressed in heart and spleen, how to exclude the effect of PD1 antibody on these tissues?

According to the images of the heart (Author response image 2 left) and spleen (Author response image 2 right) during mouse dissection, the morphology and size of the two organs were similar in HFD-CN and HFD-PD1 group. Moreover, relevant literature indicated that PD-1 blockade had little impact on the number and function of transferred T cells within the spleen(Peng et al.)，and anti-PD-1 had no effect on mouse splenic cell proliferation (Shindo et al.).Du et al. showed in their study that single use of PD-1 antibody (10 mg/kg, once every three days, for 4 weeks) did not affect mouse heart (Du et al.). Both our results and related literature indicated that PD 1 antibody should not have adverse effects on the heart and spleen.

**Author response image 2. sa3fig2:** The images of the heart and spleen of mice.

**Reviewer #2 (Public Review):**

Thank you very much for your advices and comments. We have seriously considered your suggestion and will focus on it in our future research.

Weakness1. BATF protein is also abundantly expressed in control hepatocyte, but the knockdown of BATF had no effect on lipid accumulation. Besides, the expression of BATF was elevated by high fat diets. So it will be interesting to investigate its role in the liver by using its hepatic conditional knockout mice.

We appreciate the reviewers' suggestion to investigate other functions of BATF in the liver besides its protective role in a high-fat environment. However, we did not use BATF knockout mice in this study because our data indicated that BATF knockdown had no effect on lipid accumulation. We will pursue further research and validation in future studies.

1. The data for the direct regulation of BATF on PD1 and IL-27 is not enough, it is better to carry out CHIP experiment to further confirm it.

Thank you for your valuable comments. The article by Kevin Man et al. found that, upregulation of transcription factor BATF regulates PD1 expression and repairs impaired cellular metabolism (Man et al.). This confirms that BATF has a regulatory effect on PD1. And in our manuscript, the dual luciferase reporter assay of BATF and PD1 confirmed that BATF can regulate the expression of PD1(Fig 5G). This confirms that BATF has a regulatory effect on PD1. We do not have conclusive evidence for a direct interaction between BATF and IL-27 yet, but there are some relevant studies that support their connection. For instance, BATF and IRF1 were found to be transcription factors induced early by IL-27 treatment, and essential for Tr1 cell differentiation and function, both in vitro and in vivo (Karwacz et al.). Moreover, Zhang et al. identified BATF as one of the transcription factors regulating IL-27 expression by transcription factor prediction and RNA sequencing analysis (Zhang et al.). These results lay the foundation for elucidating the regulation of PD1 and IL-27 by BATF.

**Reviewer #2 (Recommendations For The Authors):**
1. In Figure 3D, which subunit of AMPK was tested, alpha, beta or gamma?

Thank you for your valuable comments. We detected the expression level of AMPKα1, We have modified the relevant names in the figure and manuscript.

Reference：

Du, Shisuo, et al. "Pd-1 Modulates Radiation-Induced Cardiac Toxicity through Cytotoxic T Lymphocytes." 13.4 (2018): 510-20. Print.

Karwacz, Katarzyna, et al. "Critical Role of Irf1 and Batf in Forming Chromatin Landscape During Type 1 Regulatory Cell Differentiation." 18.4 (2017): 412-21. Print.

Man, Kevin, et al. "Transcription Factor Irf4 Promotes Cd8+ T Cell Exhaustion and Limits the Development of Memory-Like T Cells During Chronic Infection." 47.6 (2017): 1129-41. e5. Print.

Peng, Weiyi, et al. "Pd-1 Blockade Enhances T-Cell Migration to Tumors by Elevating Ifn-Γ InducibleChemokinespd-1 Blockade Improves the Effectiveness of Act for Cancer." 72.20 (2012): 5209-18. Print.

Shindo, Yuichiro, et al. "Interleukin 7 and Anti-Programmed Cell Death 1 Antibody Have Differing Effects to Reverse Sepsis-Induced Immunosuppression." 43.4 (2015): 334. Print.

Zhang, Huiyuan, et al. "An Il-27-Driven Transcriptional Network Identifies Regulators of Il-10 Expression across T Helper Cell Subsets." 33.8 (2020): 108433. Print.